# Importance-aware Shared Parameter Subspace Learning for Domain Incremental Learning

Shiye Wang
Beijing Institute of Technology
Beijing, China
sywang@bit.edu.cn

Changsheng Li*
Beijing Institute of Technology
Beijing, China
lcs@bit.edu.cn

Jialin Tang
Beijing Institute of Technology
Beijing, China
tjl@bit.edu.cn

Xing Gong
China academy of aerospace science
and innovation
Beijing, China
xavierxgong@gmail.com

Ye Yuan
Beijing Institute of Technology
Beijing, China
yuan-ye@bit.edu.cn

Guoren Wang
Beijing Institute of Technology
Beijing, China
wanggrbit@126.com

## Abstract

Parameter-Efficient-Tuning (PET) for pre-trained deep models (e.g., transformer) hold significant potential for domain increment learning (DIL). Recent prevailing approaches resort to prompt learning, which typically involves learning a small number of prompts for each domain to avoid the issue of catastrophic forgetting. However, previous studies have pointed out prompt-based methods are often challenging to optimize, and their performance may vary non-monotonically with trainable parameters. In contrast to previous prompt-based DIL methods, we put forward an importance-aware shared parameter subspace learning for domain incremental learning, on the basis of low-rank adaption (LoRA). Specifically, we propose to incrementally learn a domain-specific and domain-shared low-rank parameter subspace for each domain, in order to effectively decouple the parameter space and capture shared information across different domains. Meanwhile, we present a momentum update strategy for learning the domain-shared subspace, allowing for the smoothly accumulation of knowledge in the current domain while mitigating the risk of forgetting the knowledge acquired from previous domains. Moreover, given that domain-shared information might hold varying degrees of importance across different domains, we design an importance-aware mechanism that adaptively assigns an importance weight to the domain-shared subspace for the corresponding domain. Finally, we devise a cross-domain contrastive constraint to encourage domain-specific subspaces to capture distinctive information within each domain effectively, and enforce orthogonality between domain-shared and domain-specific subspaces to minimize interference between them. Extensive experiments on image domain incremental datasets demonstrate the

effectiveness of the proposed method in comparison to the related state-of-the-art methods.

## CCS Concepts

• **Computing methodologies** → **Artificial intelligence**.

## Keywords

Domain incremental learning; Parameter-efficient tuning; Domain-shared and domain-specific subspaces; Importance-aware

**ACM Reference Format:**
Shiye Wang, Changsheng Li, Jialin Tang, Xing Gong, Ye Yuan, and Guoren Wang. 2024. Importance-aware Shared Parameter Subspace Learning for Domain Incremental Learning. In *Proceedings of the 32nd ACM International Conference on Multimedia (MM '24), October 28-November 1, 2024, Melbourne, VIC, Australia.* ACM, New York, NY, USA, 10 pages. https://doi.org/10.1145/3664647.3681411

## 1 Introduction

Domain incremental learning (DIL) has attracted increasing attention in multimedia and machine learning communities [34, 47]. In DIL, it's assumed that domains are constantly evolving and often exhibit significant variation in sequence, with domain indexes being not provided for inference. Due to the dynamically changing characteristic of data distributions in DIL, traditional deep models face a significant challenge: simply fine-tuning them on new domains often results in a substantial performance drop on previous domains, a phenomenon known as "catastrophic forgetting" [29]. To this end, Parameter-Efficient-Tuning (PET) offers a promising approach for DIL, which adapts pre-trained deep models (e.g., transformer) to different domains using significantly fewer learnable parameters and resources.

In recent years, many PET approaches have been proposed for DIL [42, 47, 50], where one popular strategy is the adoption of prompt learning, an emerging paradigm derived from NLP. Prompt learning enables pre-trained language models to be repurposed for various tasks without the need for retraining. Among these prompt-based methods, they typically learn a small number of prompts for each domain to avoid the issue of catastrophic forgetting. For instance, L2P [50] is the pioneering work that establishes a prompt pool, dynamically guiding a pre-trained model to learn from domains sequentially. S-Prompts [47] attempts to independently

---

*Corresponding author

learn prompts across domains with pre-trained transformers, where S-Prompts only necessitates a single cross-entropy loss during training and a straightforward K-NN operation for domain identification during inference. CODA-Prompt [42] replaces the prompt pool with a decomposed prompt, which consists of a weighted sum of learnable prompt components. This decomposition enhances prompting capacity by expanding into a new dimension. While these methods have achieved promising performance, previous studies have pointed out that prompt-based methods are often challenging to optimize, and their performance may vary non-monotonically with trainable parameters [13].

In contrast to prompt-based DIL methods, we put forward a new PET method for DIL in this paper, inspired by the low-rank adaption (LoRA) technique. LoRA was originally proposed for large language models, which freezes the pre-trained model weights and integrates two trainable low-rank matrices into each layer of the Transformer architecture, significantly diminishing the number of trainable parameters for downstream tasks. Because of its impressive performance, LoRA has been widely applied to various tasks, such as text understanding [54], image compression [28], multi-task learning [45], etc. However, as of now, there is few study on employing LoRA for domain-incremental learning. A straightforward approach for DIL is to apply LoRA to acquire two low-rank matrices for each newly coming domain. However, such a strategy overlooks the potential to capture relations among multiple domains and uncover shared information across different domains, leading to suboptimal outcomes.

In light of these, we propose to learn an importance-aware shared parameter subspace learning for DIL. Specifically, we attempt to decouple each parameter matrix into three distinct components: a domain-specific matrix tailored to each domain, a domain-shared matrix holding relevance across all domains, and a low-rank matrix serving as the coefficient matrix. Through this approach, we can effectively capture shared information spanning diverse domains. To mitigate the issue of catastrophic forgetting, we present a momentum update strategy to learn the domain-shared subspace, facilitating the smooth accumulation of knowledge within the current domain. Moreover, considering that domain-shared information may hold varying degrees of importance to different domains (which has been verifies in Figure 2), we design a dynamic importance allocation mechanism that adaptively assigns an importance weight to the domain-shared subspace specific to each domain. In addition, we devise a cross-domain contrastive constraint to effectively encourage domain-specific subspaces to capture distinctive information within each domain. Finally, we impose another constraint ensuring orthogonality between domain-shared and domain-specific subspaces, with the goal of minimizing interference between them.

Our contributions are summarized as follows:

- We propose an shared parameter subspace learning approach for domain incremental learning, building upon LoRA. We incrementally learn a domain-specific and domain-shared low-rank parameter subspace for each domain in a momentum update manner, which can effectively capture shared information across different domains.
- We design an importance-aware mechanism to adaptively weight the domain-shared subspace for each domain, thereby

further improving the performance. Moreover, we present two constraints on domain-specific and domain-shared subspaces to enhance their representation capabilities.
- Extensive experiments on three image domain incremental datasets demonstrate the effectiveness of our proposed method in comparison with the state-of-the-art approaches.

## 2 Related Work

In this section, we review some related works, including continual learning and parameter-efficient tuning methods.

### 2.1 Continual Learning

Continual learning (CL) aims to enable the model to adapt to new class, new tasks or domains without suffering from catastrophic forgetting [38, 52, 55]. There are three common CL setups: task-incremental learning (TIL) [14, 32, 36], class incremental learning (CIL) [1, 37, 50], and domain incremental learning (DIL) [5, 40, 47]. In fact, knowing task identity during inference in TIL limits its practical utility. While in CIL, classes usually stem from the same domain, thus partially alleviating the challenge. This paper delves into DIL, where classes remain constant but domains vary significantly in sequence, without task indexes provided for inference.

In recent years, many DIL methods have been proposed, such as replay-based methods [2, 40, 44] and rehearsal-free methods [42, 47, 50]. In this work, we focus on rehearsal-free methods due to their increased practical applicability. The representative rehearsal-free methods primarily rely on prompt learning including L2P, S-Prompts, and CODA-Prompt [42, 47, 50]. These prompt-based methods fall under the umbrella of Parameter-Efficient-Tuning (PET), commonly characterized by learning a limited number of prompts for each domain. These approaches circumvent the necessity of retraining pre-trained deep models and helps alleviate the issue of catastrophic forgetting. However, as aforementioned, optimizing prompt-based methods can often be challenging, and their performance may exhibit non-monotonic variations with trainable parameters. In contrast to these prompt-based DIL methods, we attempt to learn a shared parameter subspace across various domains to capture common information among them for improving the performance of DIL.

In addition, there is a category of methods loosely related to DIL, called domain adaptation (DA) [4, 21, 25, 35]. The main difference between DA and DIL lies in their respective objectives. DA primarily aims to enhance the accuracy of target domains, whereas DIL focuses on minimizing the cumulative error across all domains, alongside implementing measures to mitigate forgetting on earlier domains. Notably, DA methods often require access to target domain data for distribution matching, making them not directly applicable to DIL.

### 2.2 Parameter-Efficient Tuning

As deep learning models continuously grow in size, it becomes crucial to study on parameter efficient methods. In recent years, there has been a significant increase in research focused on Parameter-Efficiency-Tuning (PET) [13]. The representative methods include Adapter based methods [18, 33, 43], prompt-based methods [11,

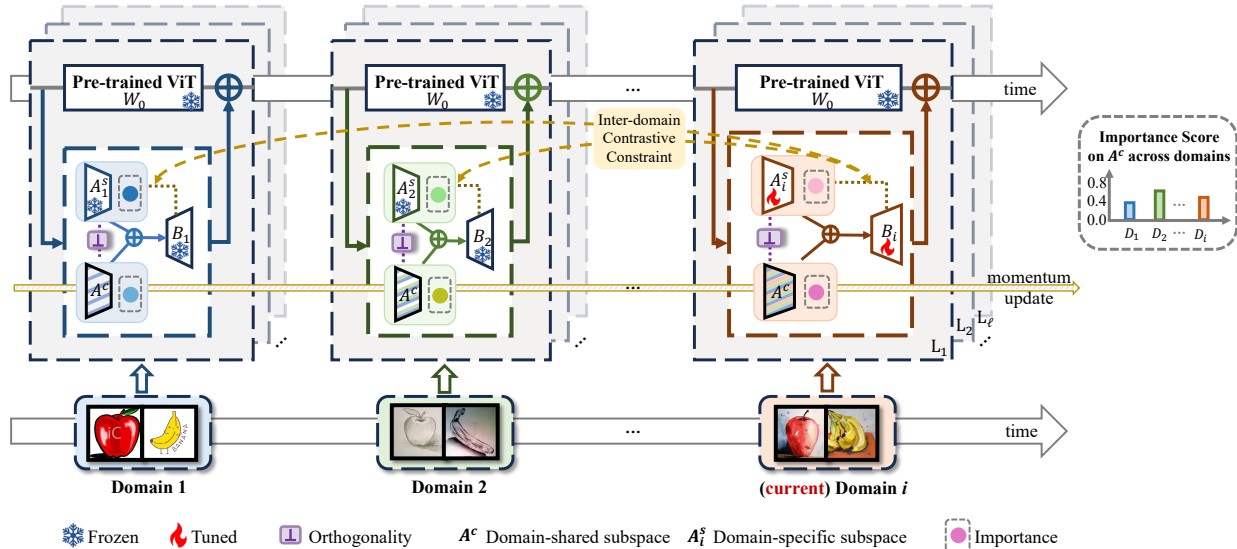

**Figure 1: Illustration of our domain incremental learning framework. To exploit the interdependencies among sequential domains, we attempt to incrementally learn a subspace $A_i^s$ specific to each newly coming domain $i$, and a subspace $A^c$ shared by all domains. Meanwhile, we present a momentum update strategy on $A^c$ to facilitate the smooth accumulation of knowledge in the current domain while mitigating the risk of forgetting knowledge acquired from previous domains. Moreover, given that $A^c$ might hold varying degrees of importance to different domains, we design an importance-aware mechanism that adaptively assigns an importance weight to $A^c$ for the corresponding domain. In addition, we impose an cross-domain contrastive constraint between the current domain-specific subspace $A_i^s$ and the historical domain-specific subspace $A_j^s$, $j < i$, as to amplify the distinctions across different domains. Finally, we enforce orthogonality between $A^c$ and each $A_i^s$ to further enhance their representation capacities.**

41, 48], and LoRA-based methods [6, 17, 31]. Adapter [12] is initially introduced to incorporate efficient and learnable modules for adapting a fixed pre-trained transformer to various new textual tasks in the NLP field. Prompt tuning [19] and Prefix tuning [22] insert learnable tokens to the input or hidden tokens, enabling the adaptation of the transformer to new tasks. LoRA [13] injects trainable low-rank adaption modules to the frozen pre-trained transformer, greatly reducing the number of trainable parameters for downstream tasks. Due to the substantial advantages of LoRA, [46] extends LoRA for language model continual learning. These PET methods have been developed for various tasks, including visual transfer learning [3, 15], video action recognition [51], image generation [27], and visual continual learning [9, 47], etc. In this study, we attempt to explore the utilization of LoRA tailored for addressing the domain-incremental learning challenge.

## 3  Proposed Method

In this section, we will elaborate our proposed method. We will begin by presenting the problem formulation and then provide a detailed introduction to our method.

### 3.1  Problem Formulation

We concentrate on continual learning with incremental domains, i.e., domain incremental learning (DIL). In DIL, a model sequentially absorbs $M$ incoming domains $\mathcal{D} := \{\mathcal{D}_1, \mathcal{D}_2, \cdots, \mathcal{D}_M\}$, where the distribution of the involved $M$ domains commonly varies a lot,

while the categories between them keep the same in $\mathcal{Y}$. The model $f(\cdot; W, \theta)$ predicts the label $\hat{y} \in \mathcal{Y}$ for a given sample $x \in \mathcal{D}_i$, where $W$ and $\theta$ denotes the feature extractor and classification head respectively, and the $i^{th}$ domain $\mathcal{D}_i$ has $N_i = |\mathcal{D}_i|$ samples. In this paper, we aim to explore LoRA-based Parameter-Efficient Tuning (PET) for DIL, where we exploit a small number of learnable parameters to effectively fine-tuning the pre-trained model. We let the feature extractor $W$ consist of the fixed pre-trained model $W_0$ and the learnable PET module $\Delta W$. Inspired by LoRA [13], we present a general objective function for DIL by minimizing the following cross-entropy loss as:

$$\mathcal{L} = \frac{1}{|\mathcal{D}_i|} \sum_{(x,y) \in \mathcal{D}_i} \mathcal{L}_{ce}(f(x, W_0; \Delta W_i, \theta_i), y), \qquad (1)$$

where $y$ is the ground truth of sample $x$ in the current domain $\mathcal{D}_i$. Note that the domain identity of samples are not provided during inference. $\theta_i$ can be optimized as most of DIL methods [42, 50]. Note that we utilize the Vision Transformer (ViT) [7] as our pre-trained model $W_0$, and we keep $W_0$ to be frozen during learning all domains. Here we can omit it for convenience.

In order to efficiently optimize $\Delta W_i$, a naive method is to directly apply LoRA to acquire two low-rank matrices for each newly coming domain. As aforementioned, this strategy fails to harness the potential of capturing cross-domain relations and discovering shared information across diverse domains, ultimately resulting in suboptimal outcomes. Therefore, our objective is to incrementally

learn shared parameter subspace to capture the common information across different domains for DIL.

## 3.2 Framework Overview

We propose an importance-aware shared parameter subspace learning network for DIL, as illustrated in Figure 1. Our method attempts to model the relations across the domains in sequence while mitigating the issue of forgetting. As depicted in Figure 1, we first attempt to decouple each parameter matrix $\Delta W_i$ into three distinct components: a domain-specific matrix $A_i^s$ tailored to each domain $i$, a domain-shared matrix $A^c$ holding relevance across all domains, and a low-rank matrix $B_i$ serving as the coefficient matrix. To ensure that the shared matrix $A^c$ can accumulate crucial common information from each domain and alleviate forgetting, we present a momentum update strategy on $A^c$ to smoothly accumulate of knowledge of all domains. Moreover, we devise an importance mechanism to weight $A^c$ for each domain, given that $A^c$ has varying degrees of importance to different domains. Moreover, we impose a cross-domain contrastive constraint on different domain-specific subspaces $A_i^s$ to capture distinctive information within each domain. Finally, we present another orthogonality constraint on domain-shared subspace $A^c$ and each domain-specific subspace $A_i^s$, so as to minimize the interference between them.

## 3.3 Shared Parameter Subspace Learning

Here we first introduce how to decouple each parameter matrix $\Delta W_i$ into three distinct components: a domain-specific matrix $A_i^s$, a domain-shared matrix $A^c$, and a low-rank coefficient matrix $B_i$.

### 3.3.1 *Parameter Subspace Decomposition for DIL.* In fact, we can view the low-rank parameter matrix $\Delta W_i \in \mathbb{R}^{d_1 \times d_2}$ for domain $i$ as an inner product between two low-rank matrices $A_i$ and $B_i$ as:

$$\Delta W_i = B_i A_i, \tag{2}$$

where $A_i \in \mathbb{R}^{r \times d_2}$ and $B_i \in \mathbb{R}^{d_1 \times r}$. $r \ll \min(d_1, d_2)$. The parameter subspace $A_i$ can be regarded as a base matrix and the parameter subspace $B_i$ can be regarded as a coefficient representation matrix. By low-rank matrix decomposition, the most significant information within $W_i$ can be captured by $A_i$ and $B_i$. $A_i$ and $B_i$ can be spanned by $r$ row vectors, respectively:

$$A_i = \text{span}\{a_i^1, \cdots, a_i^j, \cdots, a_i^r\}, \quad a_i^j \in \mathbb{R}^{d_2},$$
$$B_i = \text{span}\{b_i^1, \cdots, b_i^j, \cdots, b_i^r\}, \quad b_i^j \in \mathbb{R}^{d_1}. \tag{3}$$

However, Eq. (2) does not explicitly model the relationships among different domains, leading to suboptimal performance. To this end, we further decompose the low-rank parameter matrix $A_i$ of each domain $i$ into two components: $A_i^s$ that is specific to each domain $i$, and $A^c$ that is shared by all $M$ domains. The formulation is as follow:

$$A_i = A_i^s + A^c, \tag{4}$$

where the matrices $A_i^s$ and $A^c$ can be represented by $r$ row vectors, denoted as:

$$\begin{cases} A_i^s = \text{span}\{a_i^{s,1}, a_i^{s,2}, \cdots, a_i^{s,r}\} \\ A^c = \text{span}\{a^{c,1}, a^{c,2}, \cdots, a^{c,r}\}. \end{cases} \tag{5}$$

Therefore, given a frozen pre-trained model $W_0$, we can efficiently train the model for the current domain $i$ by:

$$W_0 + \Delta W_i = W_0 + B_i(A_i^s + A^c). \tag{6}$$

Note that Eq. (6) is applied to each layer of the pre-trained model. For the sake of convenience, we omit the index of the layer. By Eq. (6), we can decouple the parameter subspace of each domain into a domain-specific parameter subspace and a domain-shared parameter subspace.

After that, we can plug Eq. (6) into Eq. (1), and obtain a new loss function as:

$$\mathcal{L} = \frac{1}{|\mathcal{D}_i|} \sum_{(x,y) \in \mathcal{D}_i} \mathcal{L}_{ce}(f(x, W_0; B_i(A_i^s + A^c), \theta_i), y). \tag{7}$$

### 3.3.2 *Momentum Update on Domain-shared Subspace.* As domains arrive sequentially, direct optimization of the shared subspace $A^c$ for each newly arriving domain leads to updates primarily focused on learning the current domain. However, this excessive emphasis on the current domain may lead to the knowledge associated with previous domains being overwritten, potentially resulting in forgetting the knowledge of earlier domains. To alleviate this issue, we introduce a momentum update strategy to optimize the domain-shared subspace for each domain, facilitating the seamless integration of learned knowledge across all historical domains, inspired by [10]. By leveraging the momentum update strategy to optimize the shared subspace $A^c$, we can preserve the direction and velocity of updates from previous domains. This smoothing update trajectory can reduce excessive adjustments to previous domains, thus alleviating the issue of forgetting. Formally, we update $A^c$ by exploiting the smooth copies of previous domain-specific subspaces $A_i^s$ in every update step $t$ as:

$$A^c(t) \leftarrow \eta A^c(t-1) + (1-\eta)A_i^s(t-1), \tag{8}$$

where $\eta \in (0, 1)$ is a large (i.e., close to 1) momentum coefficient. Note that $A^c(t)$ is updated at every step $t$ as new domains continuously arrive. The domain-specific parameter subspace $A_i^s(t-1)$ is updated by back-propagation.

## 3.4 Importance-aware Subspace Enhancement

In this section, we will introduce how to further enhance the capacity of the decomposed parameter subspaces, so as to improve the performance of DIL.

### 3.4.1 *Dynamic Importance Allocation Mechanism.* Given that domain-shared parameter subspace $A^c$ holds varying degrees of importance to different domains (which has been verified in Figure 2), thus it is crucial to design a dynamic importance allocation mechanism to adaptively assign a weight to $A^c$ for each domain. To achieve this, we introduce a hyper-parameter $\rho_i(t)$ to dynamically weight the domain-shared parameter subspace at every training step $t$ as:

$$\Delta W_i(t) = B_i(t)(\rho_i(t)A_i^s(t) + (1-\rho_i(t))A^c(t)). \tag{9}$$

To automatically learn the hyperparameter $\rho_i(t)$, we design an importance allocation mechanism to dynamically learn $\rho_i(t)$ for each domain based on their sensitivity in the training dynamics. As pointed out in [24, 53], the sensitivity of parameters essentially approximates the parameter change in loss: If the removal of a

parameter causes a large influence on the loss, then the model is highly sensitive to the parameter, indicating that the parameter is extremely important. Motivated by this, we attempt to leverage the sensitivity of parameter subspaces to calculate the importance scores for each domain. An intuitive idea is that the higher the sensitivity of a subspace, the higher the corresponding importance score should be allocated to that subspace. Note that in order write conveniently, we will omit the symbol $t$.

To this end, we consider the average sensitivity of each parameter within subspace $A_i^s$ as the overall contribution $Q_i$ of $A_i^s$ to the model performance. To ensure comparability across $n$ subspaces for all domains, we normalize $Q = \{Q_i^1, Q_i^2, \cdots, Q_i^n\}$ of $n$ subspaces into a standard Gaussian distribution. Then we impose a sigmoid function to obtain the importance score $\rho_i \in (0, 1)$ for each subspace $A_i^s$. Therefore, we can calculate the importance score of subspaces $A_i^s$:

$$\rho_i = 1/(1 + \exp \frac{E(Q) - Q_i}{\sqrt{Var(Q)}}),$$

$$Q_i = \frac{1}{h} \sum_{j=1}^{h} S(g_i^j), \qquad (10)$$

where $h = r \times d_2$ denotes the number of parameters, and $g_i = Fla(A_i^s)$ denotes that we flatten the matrix $A_i^s$ into a long vector $g_i$. $g_i^j$ is the $j$-th element in the vector $g_i$. $S(\cdot)$ denotes a sensitivity function for each parameter. $E(Q)$ and $\sqrt{Var(Q)}$ denote the mean and standard deviation of the contribution of $n$ subspaces.

After that, we define the sensitivity of an individual parameter as the absolute value of the product between the gradient and the parameter[24, 30]. We can calculate the sensitivity of the $j^{th}$ parameter in subspace $A_i^s$ as:

$$S(g_i^j) = |g_i^j \nabla_{g_i^j} \mathcal{L}|, \qquad (11)$$

where Eq.(11) approximates the change in loss when a parameter of subspace is zeroed out. It indicates that if the exclusion of a parameter from $A_i^s$ has a significant influence, then the model is highly sensitive to that parameter. Consequently, it is crucial to assign a high level of importance to that particular parameter.

In fact, the sensitivity function $S$ calculates the loss changes reflected by an individual batch of samples. To mitigate the evaluation error caused by the sensitivity of a single batch, we can employ exponential moving average to smooth $S(g_i^j)$:

$$\bar{S}(g_i^j) \leftarrow \lambda_1 \bar{S}(g_i^j) + (1 - \lambda_1) S(g_i^j), \qquad (12)$$

where $\lambda_1 \in (0, 1)$ is the hyperparameter that controls the proportion of historical records and the current batch in the moving average.

Given the intricate training dynamics may incur high variability and significant uncertainty, we introduce an additional computation of the uncertainty for sensitivity. The uncertainty captures the local temporal changes in sensitivity and is defined as:

$$U_i^j = |S(g_i^j) - \bar{S}(g_i^j)|. \qquad (13)$$

After that, we also smooth uncertainty by exponential moving average as:

$$\bar{U}_i^j \leftarrow \lambda_2 \bar{U}_i^j + (1 - \lambda_2) U_i^j. \qquad (14)$$

Finally, we adopt the the product between the sensitivity and uncertainty as the enhanced sensitivity as:

$$S(g_i^j) \leftarrow \bar{S}(g_i^j) \cdot \bar{U}_i^j. \qquad (15)$$

We dynamically allocate importance to different parameter subspaces by calculating their respective importance scores using the enhanced sensitivity measures, as defined in Eq. (15).

### 3.4.2 Cross-domain Contrastive on Domain-specific Subspaces.
In order to enable domain-specific subspaces $A_i^s$ to learn more discriminative knowledge specific to the current domain compared to other domains, we impose an cross-domain contrastive constraint on these domain-specific subspaces, thereby amplifying the distinctions between them. The idea is as follows: we expect that the probability of the current domain's data belonging to the true class predicted by the $i^{th}$ adaptation model $f(\cdot; \Delta W_i, \theta_i)$ should be higher than the highest probability using the adaptation models $f(\cdot; \Delta W_j, \theta_j)$, where $j < i$, for each historical domain $j$. We compute the positive discrepancy probability $p_i^l$ of sample $x_l \in \mathcal{D}_i$ from the $i^{th}$ adaptation module on the correct class $y_l$ as:

$$p_i^l = 1_{c=y_l} \cdot softmax(f(x_l, W_0; \Delta W_i, \theta_i)), \qquad (16)$$

where the output of $f(\cdot; \Delta W_i, \theta_i)$ has $c$ dimensions, and $p_i^l$ denotes the probability belonging to the true class $y_l$. We get the negative discrepancy probability $p_j^l$ of sample $x_l$ from the $j^{th}$ adaptation module trained on the historical domain $j$ ($j < i$) as:

$$p_j^l = \max(softmax(f(x_l, W_0; \Delta W_j, \theta_j))). \qquad (17)$$

Then we can formulate the cross-domain contrastive loss based on these discrepancy probabilities as follow:

$$\mathcal{L}_c = -\frac{1}{|\mathcal{D}_i|} \sum_{(x_l, y_l) \in \mathcal{D}_i} \log \frac{\exp(p_i^l/\tau)}{\exp(p_i^l/\tau) + \sum_{j=1}^{i-1} \exp(p_j^l/\tau)}, \qquad (18)$$

where the $i^{th}$ domain has $|\mathcal{D}_i|$ samples in total, and $\tau$ is a temperature parameter. Note that we only update the domain-specific subspaces $A_i^s$ and its weighting coefficients $B_i$ when adapting to current domain $i$, making the historical adaption matrices $A_j^s, B_j$ for $j = 1, \cdots, i - 1$ and the domain-shared subspace $A^c$ frozen.

By the cross-domain contrastive constraint, we can encourage the model to effectively capture distinctive information specific to each domain while maintaining a clear separation between them.

### 3.4.3 Orthogonality on Parameter Subspaces.
Moreover, to minimize the interference between the domain-shared subspace and domain-specific subspaces, we propose enforcing orthogonality between each parameter vector $a_i^s$ in the domain-specific subspace $A_i^s$ and each parameter vector $a^c$ of the domain-shared subspace $A^c$. This approach reduces parameter redundancy and enhances their representation capabilities. We formulate the orthogonality on these two kinds of subspaces as:

$$< a^c, a_i^s >= 0, \quad \forall a_i^s \in A_i^s, a^c \in A^c, \qquad (19)$$

where $a_i^s, a^c \in \mathbb{R}^{d_2}$. We can achieve this by exploiting an orthogonality penalty loss as:

$$\mathcal{L}_o = ||(A_i^s)^\top * A^c||^2. \qquad (20)$$

## 3.5 Optimization

After introducing all the components of our work, we now give the final objective function based on (7), (18) and (20), which can be expressed as:

$$\mathcal{L}_{final} = \mathcal{L} + \alpha \mathcal{L}_c + \beta \mathcal{L}_o, \tag{21}$$

where $\alpha$ and $\beta$ are two trade-off hyperparameters.

During training, we optimize our model by minimizing the loss in Eq.(21) as the domains continuously arrive. We provide the overall optimization algorithm in Algorithm 1 of the Appendix A. At inference time, as the domain identifier is not available, we employ an existing method to infer the domain identifier, similar to the approach described in [47]. Specifically, we extract the features of each domain using the pre-trained model combined with the first low-rank adaptation model. Subsequently, we apply the k-means algorithm to these features to obtain a set of $k$ prototype vectors that effectively represent the information of current domain. For inference the domain identifier of the coming samples, we utilize the same pre-trained model with the first low-rank adaptation model to extract features of the input samples. We then estimate the nearest cluster center based on these features and use the index of the selected nearest cluster center as the domain identifier, denoted as $q$. Consequently, we exploit the $q^{th}$ low-rank module combined with the pre-trained model to make prediction.

## 4 Experiments

## 4.1 Experimental Settings

*4.1.1* ***Datasets***. We conduct experiments on three standard DIL benchmark datasets: CDDB [20], DomainNet [35], and CORe50 [26]. CDDB is a continual deepfake detection dataset, which aims to identify real and fake images across various domains. We choose the most challenging Hard track as in [47]. We denote it as CDDB-Hard. It involves 5 sequential deepfake detection domains with roughly 27,000 images, i.e., GauGAN, BigGAN, WildDeepfake, WhichFaceReal, and SAN. DomainNet has 6 distinct domains with roughly 600,000 images, and each domain containing 345 categories. We adopt the same DIL setup as DomainNet in [47]. CORe50 is a publicly available dataset for continual object recognition. It consists of 11 different domains, where each domain contains 50 categories. We follow [8, 50] to utilize 8 domains with 120,000 images for incremental training, and leave the rest 3 unseen domains for testing.

For CDDB-Hard, we report the average forward detection accuracy across all domains, similar to [20]. For DomainNet and CORe50, we report the average forward classification accuracy of all domains, similar to [47] and [50].

*4.1.2* ***Baselines***. Since our method belongs to the category of rehearsal-free domain incremental learning (DIL) approaches, we compare our method with three state-of-the-art rehearsal-free DIL methods, i.e., prompt-based methods including L2P [50], S-Prompts [47], and CODA-Prompt [42]. In addition, we compare our method with two representative parameter-efficient-tuning (PET) methods for continual learning, including one prompt-based method, DualPrompt [49] and a hybrid PET method, LAE [9]. Moreover, we also take one regularization-based method, EWC [16], and one distillation-based method, LwF [23], as our baselines. Finally, we

apply LoRA to each domain, and acquire the low-rank matrices independently across different domains, denoted as DiLoRA. We take DiLoRA as another baseline. To ensure a fair comparison, we evaluate all PET based methods using the same model, ViT-B/16 [7], pre-trained on the ImageNet21k [39] dataset.

*4.1.3* ***Implementation Details***. We conduct experiments using the GeForce RTX 3090 Ti GPU. Input images are randomly resized to $224 \times 224$ resolution and normalized by standard deviation, following S-Prompts [47]. All baselines employ the same data augmentation strategy as ours for fair comparisons. Optimization across all experiments is performed using the Adam optimizer. We set the learning rate to 0.003 and utilize the batch size of 50 for three datasets. For the CDDB-Hard dataset, we set the hyper-parameter $\alpha$ to 0.5 and the hyper-parameter $\beta$ to 0.5. For the DomainNet dataset, we set $\alpha$ to 0.1 and $\beta$ to 0.5. For the CORe50 dataset, we set $\alpha$ to 4 and set $\beta$ to 2. We train each domain 13 epochs, 5 epochs, and 3 epochs for CDDB-Hard, DomainNet, and CORe50, respectively. For the three datasets, we set the exponential moving average hyperparameters $\lambda_1$ and $\lambda_2$ to 0.85 throughout the experiment. The momentum coefficient $\eta$ mentioned in Eq.(8) is empirically set to 0.9999. During inference, we set the number of clusters to $k = 5$, as recommended by S-Prompts [47]. We repeat every experiment three times to report the average result.

## 4.2 Experimental Results

*4.2.1* ***General Performance***. We conduct experiments on the three datasets, and test two kinds of different numbers of domains for each dataset: the first three domains and all five domains for CDDB-Hard, the first three domains and all six domains for DomainNet, and the first four domains and all eight domains for CORe50. The average results are reported in Table 1. It can be observed that our method consistently outperformed all other competing approaches. Particularly, our method outperforms all four Prompting-based CL methods, demonstrating the effectiveness of our parameter subspace learning approach for DIL. when compared with DiLoRA, our method achieves better performance. This demonstrates that by leveraging the shared information among different domains, we are able to enhance the overall performance of DIL. Moreover, our method outperforms LAE in a large margin. This may be because LAE aims to equally accumulate historical knowledge among all past domains. However, not all the information in these domains is helpful. Thus, it is necessary to learn importance-aware parameter subspace for DIL.

*4.2.2* ***Ablation Study***. We design several variants of our method to analyse the effect of different components. The results are listed in Table 2. "Ours-w.o.-$A^c$" and "Ours-w.o.-$A_i^s$" refer to our method without utilizing the domain-shared subspace and domain-specific subspaces, respectively. "Ours-w.o.-importance" denotes the equal contribution between domain-shared and domain-specific subspaces for each domain. Our method achieves better performance than "Ours-w.o.-$A^c$", illustrating that it is necessary to capture common information among different domains for DIL. Our method performs better than "Ours-w.o.-importance", indicating that it is important to assign different weights to the domain-shared parameter

**Table 1: Average accuracies (%) of different methods on the CDDB-Hard, DomainNet, and CORe50 datasets.**

| Dataset | CDDB-Hard | | DomainNet | | CORe50 | |
|---|---|---|---|---|---|---|
| Method | 3domains | 5domains | 3domains | 6domains | 4domains | 8domains |
| EWC [16] | 48.62±0.57 | 50.59±0.49 | 46.02±0.39 | 47.62±0.43 | 73.67±0.61 | 74.82±0.60 |
| LwF [23] | 57.74±0.50 | 60.94±0.43 | 47.35±0.34 | 49.19±0.35 | 74.73±0.38 | 75.45±0.40 |
| L2P [50] | 62.27±0.34 | 61.28±0.41 | 40.43±0.12 | 40.15±0.11 | 78.54±0.23 | 78.33±0.06 |
| DualPrompt [49] | 68.83±0.40 | 70.38±0.39 | 56.83±0.07 | 59.57±0.08 | 87.12±0.51 | 89.37±0.49 |
| S-Prompts [47] | 73.44±0.38 | 74.51±0.40 | 50.21±0.10 | 50.62±0.09 | 81.04±0.47 | 83.13±0.51 |
| CODA-Prompt [42] | 79.35±0.30 | 78.42±0.29 | 61.15±0.06 | 61.69±0.06 | 88.37±0.48 | 90.02±0.45 |
| LAE [9] | 75.94±0.44 | 79.19±0.47 | 54.86±0.20 | 53.93±0.54 | 85.68±0.54 | 89.33±0.22 |
| DiLoRA [13] | 86.37±0.37 | 86.77±0.33 | 60.32±0.11 | 64.03±0.23 | 84.97±0.50 | 86.94±0.51 |
| Ours | **91.03**±0.35 | **90.10**±0.38 | **63.14**±0.07 | **67.80**±0.11 | **89.79**±0.43 | **91.07**±0.52 |

**Table 2: Ablation study on our core components.**

| Method | CDDB-Hard | DomainNet | CORe50 |
|---|---|---|---|
| Ours-w.o.-$A^c$ | 86.77±0.33 | 64.03±0.23 | 86.94±0.51 |
| Ours-w.o.-$A_i^s$ | 66.95±0.32 | 41.02±0.09 | 84.53±0.49 |
| Ours-w.o.-importance | 88.09±0.35 | 65.65±0.11 | 88.92±0.50 |
| Ours | **90.10**±0.38 | **67.80**±0.11 | **91.07**±0.52 |

**Table 3: Ablation study on our loss function.**

| Method | CDDB-Hard | DomainNet | CORe50 |
|---|---|---|---|
| Ours-w.-cls-only | 87.72±0.31 | 64.60±0.09 | 87.43±0.50 |
| Ours-w.o.-c | 88.27±0.37 | 66.59 ±0.10 | 87.68±0.45 |
| Ours-w.o.-o | 88.94±0.41 | 66.87±0.11 | 89.71±0.47 |
| Ours | **90.10**±0.38 | **67.80**±0.11 | **91.07**±0.52 |

subspace for each domain. Moreover, when removing the domain-specific subspace for each domain, "Ours-w.o.-$A_i^s$" achieves worse performance, demonstrating domain-specific information is very useful for representing each domain.

We also verify the loss terms in our final objective function in (21). We first perform the experiments by removing both the $\mathcal{L}_o$ term and the $\mathcal{L}_c$ loss term, denoted as "Ours-w.-cls-only". "Ours-w.o.-o" means that we remove the orthogonality constraint on the domain-shared and domain-specific subspaces by setting the hyperparameter $\beta = 0$. Moreover, we test the effectiveness of the cross-domain contrastive loss by setting $\alpha = 0$, denoted as "Ours-w.o.-c". The results are listed in Table 3. It is evident that each loss term in our method contributes to the final performance of our method. More ablation studies can be found in Appendix B.1.

*4.2.3 Effect of Different Importance Weighting.* We conduct an experiment on the CDDB-Hard dataset to illustrate that domain-shared parameter subspace holds varying degrees of importance to different domains. Specifically, we use the first loss term $\mathcal{L}$ in (21) to acquire domain-shared subspace and domain-specific subspaces in a momentum update manner. After that, we manually introduce

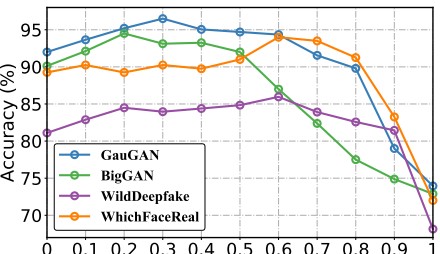

**Figure 2: Empirical study on the importance of domain-shared information for different domains. Different importance proportions of the domain-shared subspace are manually applied to four domain data, namely GauGAN, BigGAN, WildDeepfake, WhichFaceReal.**

varying proportions of importance on domain-shared subspace and domain-specific subspaces, ranging from $\{0, 0.1, 0.2, \cdots, 1\}$. The results are shown in Figure 2. As illustrated in Figure 2, we can make the following observations: (1) There is a notable variation in the importance of domain-shared information across each domain. (2) The utilization of domain-shared information enhances performance for each domain in the CDDB-Hard dataset (When the proportion is set to zero, it means that we do not use domain-shared information). (3) The optimal proportion of domain-shared information significantly varies across different domains. Therefore, we can draw the conclusion that dynamically assigning an importance weight to the domain-shared subspace benefits the final performance of DIL.

*4.2.4 Effect of the Varying Number of Vectors in Subspace.* Recall that our domain-shared and domain-specific subspaces consist of $r$ row vectors when decomposing the parameter matrix. Here we investigate the effect of the varying number of vectors on the three datasets. As illustrated in Figure 3 (a), we find that for the CORe50 dataset, setting the number of vectors within the subspace to 2 is sufficient to achieve good results. The performance of having a larger number of vectors drop slightly. This can be attributed to the relatively simple content in the image, where lower rank matrices can capture sufficient information. For the Domain-Net and CDDB-Hard datasets, we observe that setting a relatively

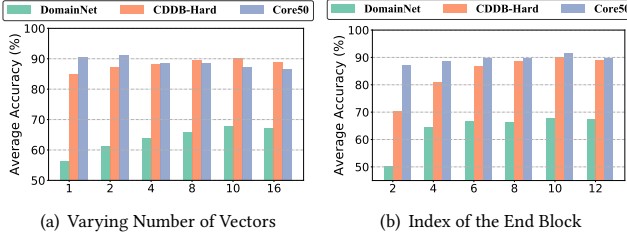

(a) Varying Number of Vectors    (b) Index of the End Block

**Figure 3: (a) The performance of the varying number of vectors within domain-common and domain-specific matrices and (b) The performance of varying the number of inserted blocks. There are 12 transformer blocks in the pre-trained model. Our decomposed subspaces are inserted into the transformer starting from the first block.**

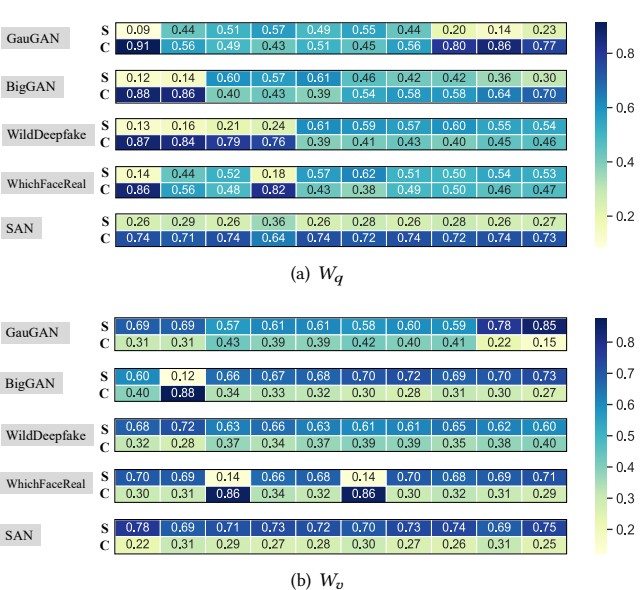

**Figure 4: Heatmap of the importance of domain-specific subspaces (denoted as S) and the importance of domain-shared subspaces (denoted as C) of five domains from CDDB-Hard.**

larger value (e.g., 10) can help the model capture more information, thereby improving its performance.

*4.2.5 **Effect of Attach Position of the Subspaces**.* Moreover, we further examine the effect of different attach positions of the decomposed subspaces to the pre-trained model. We conduct these experiments on the CDDB-Hard, DomainNet and Core50 datasets. The results have been demonstrated in Figure 3 (b). Overall, the performance of all incremental domains initially increases as the number of the inserted blocks increases, and then levels off. We observe that by attaching the decomposed subspaces to the first 10 transformer blocks, we can achieve the best performance for all the three datasets.

*4.2.6 **Visualization of the Obtained Importance**.* We visualize the obtained importance of domain-shared and domain-specific

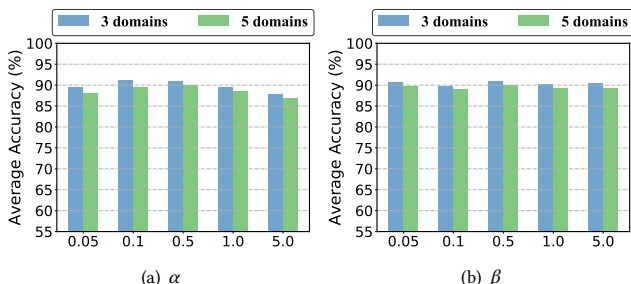

(a) $\alpha$    (b) $\beta$

**Figure 5: Parameter sensitivity analysis of the hyperparameter of the objective in Eq.(21) on CDDB-Hard.**

subspaces by our method on the CDDB-Hard dataset, as shown in Figure 4. We attach the decomposed subspaces to the first 10 transformer blocks during the incremental training process of the five domain datasets. Specifically, we introduce the decomposed subspaces into the widely used query and value projection matrix (denoted as $W_q$ and $W_v$) for the selected transformer layer, following [13]. Once the training is completed, we obtain the dynamical importance scores for different layers of all five domains. As shown in Figure 4 (a)(b), we can clearly observe that there indeed exists varying importance degrees of domain-shared subspaces across different domains. Additional visualization of the obtained importance matrices on the DomainNet and Core50 datasets can be found in Appendix B.2.

*4.2.7 **Impact for Hyper-parameters**.* Finally, we analyze the impact of the involved main hyper-parameters in our method on the performance. We examine the parameter sensitivity of $\alpha$ and $\beta$ introduced in Eq.(21) on the CDDB-Hard dataset. For evaluation, we keep all other hyperparameters to be fixed except for the one being tested. We analyse $\alpha$ and $\beta$ varying from $\{0.05, 0.1, 0.5, 1, 5\}$ respectively, and report the results in Figure 5(a) and (b). we can see the performance of our method is relatively stable in a relatively wide range. Thus it is easy to set in practical applications. More analysis on the hyper-parameter sensitivity can be found in Appendix B.3.

## 5 Conclusion

In this paper, we proposed an importance-aware shared parameter subspace learning framework for domain incremental learning (DIL). Our approach focused on incrementally learning a domain-shared low-rank parameter subspace across all domains and a domain-shared low-rank parameter subspace for each domain. Moreover, we presented a momentum update strategy to learn the domain-shared subspace, which can smoothly capture shared information across different incremental domains to mitigate forgetting. Recognizing that the importance of the domain-shared subspace may vary across domains, we proposed an importance-aware mechanism to adaptively learn importance weight for each domain. Furthermore, we imposed two constraints on the domain-specific and domain-shared subspaces, so as to enhance their representation capabilities. Extensive experiments on three publicly available DIL datasets verified the effectiveness of the proposed method, compared to the state-of-the-arts.

# Acknowledgments

This work was supported by the National Natural Science Foundation of China (NSFC) under Grant 62122013 and Grant U2001211. This work was also supported by the Innovative Development Joint Fund Key Projects of Shandong NSF under Grant ZR2022LZH007.

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
