# OpenReview forum: "Importance-aware Shared Parameter Subspace Learning for Domain Incremental Learning"
_acmmm.org/ACMMM/2024/Conference — MM2024 Poster_

### Official Review · Reviewer_GdZ9 · 2024-04-29

**Rating:** 4
**Confidence:** 2

**Summary:**

In this paper, the authors proposed an importance-aware shared parameter subspace learning for domain incremental learning, on the basis of low-rank adaption (LoRA), which incrementally learn a domain-specific and domain-shared low-rank parameter subspace for each domain, in order to effectively decouple the parameter space and capture shared information across different domains. Moreover, given that domain-shared information might hold varying degrees of importance across different domains, the authors designed an importance-aware mechanism that adaptively assigns an importance weight to the domain-shared subspace for the corresponding domain. Experiments on image domain incremental datasets demonstrate the effectiveness of the proposed method.

**Strengths:**

1. The various modules in the entire method are relatively mature. The authors designed an importance-aware mechanism that adaptively assigns an importance weight to the domain-shared subspace
for the corresponding domain.

**Limitations:**

1. The motivation discussed in the paper is not very clear. The authors claim that: 'incrementally learn a domain-specific and domain-shared low-rank parameter subspace for each domain in a momentum update manner, which can effectively capture shared information across different domains.'. However, the theoretical justifications lack. Could the authors elaborate on the theoretical justification for the effectiveness, especially regarding the role of shared information across different domains?

2. Emprical results mainly focus on visual classification tasks, are there more results on other visual tasks?

**Suitability:**

2

---

### Official Review · Reviewer_Y1rL · 2024-05-04

**Rating:** 4
**Confidence:** 3

**Summary:**

This papre puts forward an importance-aware shared parameter subspace learning for domain incremental learning, on the basis of low-rank adaption (LoRA). It focused on incrementally learning a domainshared low-rank parameter subspace across all domains and a domain-shared low-rank parameter subspace for each domain.  Extensive experiments on image domain incremental datasets demonstrate the effectiveness of the proposed method.

**Strengths:**

1. The writting is clear.
2. The subspaces as special domain and shared domain are reasonable, and an interesting idea.
3. The proposed importance allocation mechanism is to dynamically learn the hyperparameter in each domain, which is soundable.

**Limitations:**

1. The reason of adopting Lora is not clear,.
2.  Some formula symbols are not explained clearly, like (11) the loss function L is L_final in (15)?
3. Since too many parameters in this paper, it is better to annotate clear which one is learnable.

**Suitability:**

3

---

### Official Review · Reviewer_Uds4 · 2024-05-07

**Rating:** 3
**Confidence:** 2

**Summary:**

The authors propose a novel parameter-efficient tuning (PET) approach for domain incremental learning (DIL) based on low-rank adaptation (LoRA). The key contributions include learning domain-specific and shared low-rank subspaces, using momentum updates and importance weighting for the shared subspace, and imposing contrastive and orthogonality constraints. Experiments on three DIL datasets show improvements over several state-of-the-art baselines.

**Strengths:**

1. The paper addresses the challenging and practically relevant problem of domain incremental learning using a novel LoRA-based PET approach. This is a timely contribution given the growing interest in parameter-efficient tuning methods. The technical approach is well-motivated and grounded in sound principles.
2. The experimental evaluation is quite comprehensive, covering three diverse DIL datasets and comparing against a good selection of 8 recent state-of-the-art baselines.

**Limitations:**

1. While the overall results are strong, the performance gains are not uniform across the datasets. On DomainNet and CORe50, the improvements over some baselines are quite marginal.
2. The approach introduces several hyperparameters ($\alpha$, $\beta$, $r$, momentum coefficient, block insertion strategy, etc.). While the sensitivity to some of these is analyzed, it's unclear how a practitioner should set these for a new dataset.
3. The finding that the optimal importance of shared information varies widely across domains is intriguing but not sufficiently explained. What characteristics of the domains cause this? How does the proposed importance weighting mechanism capture this?
4. There are some missing details that would aid in reproducibility, such as the specific rank $r$ used for the subspaces and the layers where they are inserted.

**Suitability:**

2

---

### Official Review · Reviewer_jcuM · 2024-05-25

**Rating:** 5
**Confidence:** 3

**Summary:**

In this paper, the authors proposed an importance-aware shared parameter subspace learning framework for domain incremental learning
(DIL), which focused on incrementally learning a domain-shared low-rank parameter subspace across all domains and a
domain-shared low-rank parameter subspace for each domain. Extensive experiments validate the effectiveness of the proposed method.

**Strengths:**

1. Novel method and interesting idea. The idea of decomposing the LoRA-like parameter into domain-shared part and domain-specific part is novel and inspiring to me.
2. Extensive experiments validate the effectiveness of the proposed method.
3. The paper is well-written and easy to follow.

**Limitations:**

1. Could the proposed method be integrated with CNNs？
2. Could the method be further extended to variants of LoRA (i.e. ConvLoRA[1])?

It would be ok to answer 'No', overall, I'm satisfied with the paper.

[1] CONVOLUTION MEETS LORA: PARAMETER EFFICIENT FINETUNING FOR SEGMENT ANYTHING MODEL, ICLR 2024

**Suitability:**

2

---

### Meta-Review · Area_Chair_U1A4 · 2024-06-30

**Recommendation:** Accept (Poster)
**Confidence:** 5

**Metareview:**

This paper proposed a novel method to efficiently fine-tuning model parameters in domain incremental task. The proposed framework tries to decouple the learnable parameters with regard to being domain-specific or domain shared parameters; Thus enhancing the overall performance.

We thank the authors for their great work and the comprehensive rebuttal discussion provided. We think this work is impactful and beneficial and definitely suitable for ACM MM.

The paper addresses an important aspect. The experiments and the results are solid. The novelty of the method is original. We think this paper is useful and impactful.